# Effects of Three Lipidated Oxytocin Analogs on Behavioral Deficits in CD38 Knockout Mice

**DOI:** 10.3390/brainsci7100132

**Published:** 2017-10-16

**Authors:** Stanislav M. Cherepanov, Shirin Akther, Tomoko Nishimura, Anna A. Shabalova, Akira Mizuno, Wataru Ichinose, Satoshi Shuto, Yasuhiko Yamamoto, Shigeru Yokoyama, Haruhiro Higashida

**Affiliations:** 1Department of Basic Research on Social Recognition, Research Center for Child Mental Development, Kanazawa University, Kanazawa 920-8640, Japan; akthershirin1182@yahoo.com (S.A.); tomoko.n@hama-med.ac.jp (T.N.); ashabalova@icloud.com (A.A.S.); shigeruy@med.kanazawa-u.ac.jp (S.Y.); haruhiro@med.kanazawa-u.ac.jp (H.H.); 2Faculty of Pharmaceutical Sciences, Center for Research and Education on Drug Discovery, Hokkaido University, Kita-12, Nishi-6, Kita-ku, Sapporo 060-0812, Japan; nrh23255@gmail.com (A.M.); wichinose@pharm.hokudai.ac.jp (W.I.); shu@pharm.hokudai.ac.jp (S.S.); 3Departments of Biochemistry and Molecular Vascular Biology, Graduate School of Medical Sciences, Kanazawa University, Kanazawa 920-8640, Japan; yasuyama@med.kanazawa-u.ac.jp

**Keywords:** oxytocin, autism, CD38, lipidation, social behaviors

## Abstract

Oxytocin (OT) is a nonapeptide that plays an important role in social behavior. Nasal administration of OT has been shown to improve trust in healthy humans and social interaction in autistic subjects. As is consistent with the nature of a peptide, OT has some unfavorable characteristics: it has a short half-life in plasma and shows poor permeability across the blood-brain barrier. Analogs with long-lasting effects may overcome these drawbacks. To this end, we have synthesized three analogs: lipo-oxytocin-1 (LOT-1), in which two palmitoyl groups are conjugated to the cysteine and tyrosine residues, lipo-oxytocin-2 (LOT-2) and lipo-oxytocin-3 (LOT-3), which include one palmitoyl group conjugated at the cysteine or tyrosine residue, respectively. The following behavioral deficits were observed in CD38 knockout (CD38^−/−^) mice: a lack of paternal nurturing in CD38^−/−^ sires, decreased ability for social recognition, and decreased sucrose consumption. OT demonstrated the ability to recover these disturbances to the level of wild-type mice for 30 min after injection. LOT-2 and LOT-3 partially recovered the behaviors for a short period. Conversely, LOT-1 restored the behavioral parameters, not for 30 min, but for 24 h. These data suggest that the lipidation of OT has some therapeutic benefits, and LOT-1 would be most useful because of its long-last activity.

## 1. Introduction

Autism spectrum disorder (ASD) is a set of neurodevelopmental conditions characterized by impairments in social communication, social interactions and the presence of repetitive behaviors and restricted interests [1]. There are dozens of potentially effective intervening approaches for ASD symptoms and psychoeducational treatment [2,3], including training of emotion recognition [4]; however, a common therapeutic strategy by medicine for social impairments in patients with ASDs is still absent [5].

A great deal of research has focused on a neuroendocrine hormone, oxytocin (OT) [6,7]. This cyclic nonapeptide is mainly synthesized in the neurons of the supraoptic nucleus and paraventicular nucleus in the hypothalamus [8]. OT is secreted into the blood circulation from the posterior pituitary lobe and acts as a hormone in peripheral organs [9,10]. Recently, however, it is known that OT is a key neuromodulator in the brain for pair bonding, social behavior, social recognition and memory among different species of mammals, including humans [11,12,13,14,15,16]. Clinical trials have demonstrated a role for OT in the social behavior of humans [17,18,19]. OT has been shown to modulate specific parameters, such as improving feelings of trust [20], enhancing of the emotion recognition of facial expressions [21] and predicting mother-to-infant gaze [22]. OT increases bodily emotion recognition [23] and the tendency for anthropomorphizing [24]. A single or repetitive intranasal administration of OT in randomized controlled trials has been demonstrated to elicit the recovery of some parameters of social behavior in ASD patients [25,26], while other trials failed to show positive effects [27,28,29]. One of the possible reasons for such results for OT seems to be the pharmacokinetic properties of OT, which are similar to other natural peptides, i.e., a short lifetime in the blood [30] and a poor brain delivery profile [31].

One of the successful methods for extending the half-life of peptides in the blood is lipidation, i.e., the conjugation of a given peptide with long fatty acids [32]. Palmitoylation is one method for the lipidation of a molecule. Recently, the beneficial effects of the palmitoylation of a parental peptide molecule have been shown [33]. Though the effects of the lipidation of a parent molecule on blood-brain barrier (BBB) penetration have not been documented, hydrophobic small molecules with higher log-P values show better penetration across the BBB, compared to their parental hydrophilic congeners [34]. On the basis of these previous findings, we hypothesized that the lipidation strategy would enhance the hydrophobicity of OT and, thereby, extend its half-life in the blood and permeability through the BBB. Previously, we synthesized an OT analog, lipo-oxytocin-1 (LOT-1), in which two palmitoyl groups are conjugated at the amino group of the Cys^1^ residue and the phenolic hydroxyl group of the Tyr^2^ residue [35]. In addition, lipo-oxytocin-2 (LOT-2) and lipo-oxytocin-3 (LOT-3) were synthetized and feature the conjugation of only one palmitoyl group at the amino group of the Cys^1^ residue or the phenolic hydroxyl group of the Tyr^2^ residue, respectively [36]. LOT-1 was tested in CD157 knockout mice, an established model of the non-motor symptoms of Parkinson disease and of social avoidance in ASD [37]. A single intraperitoneal (i.p.) injection of LOT-1 rescued social anxiety and avoidance for a mouse (social target) in the open field. Expectedly, the effect of LOT-1 was shown to be stable, and it was maintained for 24 h after injection in CD157 knockout mice [35].

Recently, we also showed that the administration of LOT-1 to CD38 knockout (CD38KO or CD38^−/−^) male mice displayed a significant effect from 3 to 24 h after injection, during which OT completely lost its effect, while there was no immediate effect at 30 min after injection [36]. We speculated that LOT-1 has a long-lasting effect and is possibly metabolized to OT in the blood or brain gradually. This estimation is based on the fact that OT concentrations in the plasma and cerebrospinal fluid (CSF) after LOT-1 injection were constant at 30 min but increased at 24 h after i.p. injection [36]. LOT-2 and LOT-3 improve latency to retrieve pups to the nest on 30 min, and only LOT-2 did during 3–12 h. Nevertheless both analogs showed no effect on another parameter of parental behavior, such as time to complete retrieval of pups to the nest in 3–12 h’ time points LOT-1, however, displayed improvement in both parameters [36]. Such ineffectiveness of LOT-1 in the short latency can be explained, in part, by no direct reaction of stimulating OT receptors, in that no elevation in intracellular free calcium concentrations was found in cultured cells expressing human OT receptors stimulated by the LOT-1. On the other hand, LOT-2 showed strong acute in vivo effects and week partial agonistic effects on an elevation of calcium concentrations in vitro [36].

Here, we aimed to further confirm the prominent effects of the three OT analogs in CD38^−/−^ mice using different parameters reflecting social behavioral defects [38]. Since some behaviors have not been examined previously, we tested: disturbance in social memory and recognition in CD38^−/−^ mice [39,40], high locomotion activity in the pups [41,42], lower reward tendency examined under the sucrose preference paradigm and depression-like behavioral defects, which may be related to autistic behavior [8,35]. In addition, we judged the effectiveness of LOTs by “parental score”, rather than latency to retrieve the pups and the time to complete retrieval of the pups to the nest, as the parameter for parental behavior [36] because this behavioral score may reflect the quality of parental behavior, especially in a case of incomplete care.

## 2. Materials and Methods

### 2.1. Synthesis of LOTs

The synthesis of LOT-1, LOT-2 and LOT-3 was described previously [35,36].

### 2.2. Animals

Male and female Slc:ICR mice (Institute of Cancer Research of the Charles River Laboratories, Inc., Wilmington, MA, USA) were obtained from Japan SLC, Inc. (Hamamatsu, Japan) through a local distributor (Sankyo Laboratory Service Corporation, Toyama, Japan). The procedure to produce the CD38^−/−^ mice was described previously [43]. The offspring of wild-type (CD38^+/+^) and CD38^−/−^ mice were born in our laboratory colony. The pups were weaned at 21–28 days of age and housed in same-sex groups of five animals until pairing. A male and female of each genotype were paired and housed in a nursing cage in our laboratory under standard conditions (24 °C; 12-h light/dark cycle, lights on at 08:00) with food and water provided *ad libitum*. All animal experiments were performed in accordance with the Fundamental Guidelines for the Proper Conduct of Animal Experiments and Related Activities in Academic Research Institutions under the jurisdiction of the Ministry of Education, Culture, Sports, Science, and Technology of Japan and were approved by the Committee on Animal Experimentation of Kanazawa University (Ethics Approval Code AP-173824).

### 2.3. Parental Retrieval Test

The design of the experiments for parental retrieval behavior was described previously [36,44]. Virgin males and females of identical genotypes were paired at 56–64 days. A single male and a single female were continuously housed together in a standard mouse maternity cage from the mating period to the delivery of pups and then to postnatal days 3–5. All family units consisted of a new sire and dam, and their first litter of each genotype was used. All mice were experimentally naive. Thirty minutes before starting the experiment, the cages with the families were placed in the experimental room for habituation. The sire and dam were placed in a new clean cage with new woodchip bedding for 10 min, while the pups were left in the nest in the original cage. Five pups were randomly selected from the litter and placed individually at a site remote from the nest in the original cage. The sires and dams were returned to the original home cage in the presence of their five biological pups to assess parental behavior.

Parental retrieval behavior was scored by observing the parent behavior for 10 min following the reunion. Each sire or dam received 2 points for each pup returned completely to the nest, 1 point if the parent had contact with the pup or moved the pup to another place, and 0 if the pup was intact. After finishing of one session, the next sessions were repeated after a 10-min reset period between each session (Figure 1). In the second series of experiments, the sire received a single intraperitoneal injection of 0.3 mL of phosphate-buffered saline (PBS) or 0.3 mL of OT (1 mL per 100 g of body weight), LOT-1, LOT-2, or LOT-3 at a concentration of 100 ng/mL dissolved in PBS. Thirty minutes or 24 h after the injection, paternal retrieval behavior was examined only once, as shown in Figure 2. The behavioral tests were carried out in a randomly mixed sequence of the experimental groups. The experiments were usually performed between 10:00 and 15:00.

### 2.4. Rapid Test of Social Acquisition and Recognition

The protocol for the social acquisition and recognition test was described previously [45]; however, the experiments in the present study included a male intruder. An adult male mouse was investigated while a male intruder mouse was placed in his home cage for 1 min. The inter-trial interval was 10 min. A single-housed virgin male confronted a conspecific single-housed virgin male. When the same male was presented in four successive trials, the last trial was run to assess a new stimulus male (fifth trial). Sessions were recorded on video and scored by two observes who were not involved in the animal experiments. The duration of olfactory investigatory behaviors, including anogenital sniffing, body and tail sniffing, and head sniffing, were calculated as described [45]. The data are shown by duration and ratios between the first and fourth or fourth and fifth sessions.

### 2.5. Tail Suspension Test

The tail suspension test method was previously described [46,47]. After 30 min of habituation in the experimental room, the mice was injected i.p. with 0.3 mL of PBS or 0.3 mL of OT, LOT-1, LOT-2, or LOT-3 at a concentration of 100 ng/mL dissolved in PBS. Thirty minutes after the injection, the mice were hanged by fixing their tails by tape to the suspension bar in a plastic suspension box (55 cm height × 60 cm width × 11.5 cm depth). To prevent observing or interacting with each mouse, the mouse was separated with walls but was not able to contact or touch to the walls. Behavior was recorded for 6 min on video-source and was analyzed using ANY-Maze behavioral tracking software (Stoelting Co., Wood Dale, IL, USA). Total immobility time during the last 4 of the 6 min was used.

### 2.6. Elevated plus Maze

This test was based on a method that was described previously [48,49]. The maze was made of four black plexiglass arms with two open arms (67 × 7 cm) and two walled arms (67 × 7 × 17 cm) facing each other and connected by a neutral space in the center at 55 cm above the floor under dimly illuminated light (20 lux) [48,49]. The time spent and the frequency of entry into each arm in 8 min was automatically tracked by the camera-assisted ANY-Maze software.

### 2.7. Sucrose Preference

Experimentally naïve, young adult, male and female mice (6–8 weeks old) were given a two-bottle choice between distilled water and sucrose solution at a 1% concentration based on our previous results [50], which were both available *ad libitum*. The bottle positions remained constant. Fresh sucrose solution was prepared each day. Cumulative water and sucrose intakes during 24 h were calculated by weighing. Food was provided *ad libitum*, but food intake was not recorded in this experiment.

### 2.8. Pup Locomotion Test

Seven-day-old male pups were isolated from their parents in their home cage. Each pup was placed into the recording chamber (22 cm × 22 cm) for 3 min, and the number of grid crossings (4.5 cm × 4.5 cm) was counted [41].

### 2.9. Statistical Analysis

The two-tailed Student’s *t*-test was used for single comparisons between two groups. The rest of the data were analyzed by one-way or two-way analyses of variance (ANOVA) for two or three components, respectively. Post hoc comparisons were performed only when the main effect showed statistical significance. The *p*-values of the multiple comparisons were adjusted using Bonferroni’s correction. All data are shown as the mean ± standard error of the mean. In all analyses, *p* < 0.05 indicated statistical significance. All analyses were performed using GraphPad Prism 6 software (GraphPad Software, La Jolla, CA, USA).

## 3. Results

### 3.1. Parental Retrieval Test

We first examined parental (retrieval) behavior by sires and dams for 5 pups of wild-type (CD38^+/+^) and CD38 KO (CD38^−/−^) strains, essentially as described previously [38,44]. CD38^+/+^ primiparous dams displayed nearly complete maternal (retrieval) behavior, with scores of 9.5 ± 1.1 in the first trial and 10.0 in 2nd and 3rd trials (*n* = 10) (the ceiling value of 10) (Figure 1A). The maternal behavior by CD38^−/−^ primiparous dams exhibited a slightly low value of 6.0 ± 1.4, 7.6 ± 1.3 and 8.2 ± 0.9 for the first, second and third trials (*n* = 10), respectively, which were not significantly lower than those of the wild-type dams (Figure 1A).

Sires of the ICR strain have been reported to display parental behavior after separation with only their mate dams in a novel cage for 10 min [44]. CD38^+/+^ first-time sires displayed retrieval behavior, with scores of 6.1 ± 0.9, 6.6 ± 1.0 and 7.1 ± 1.2 (*n* = 10) in the 1st to 3rd trials (ceiling value approximately 6–7 [44]; Figure 1B), respectively. CD38^−/−^ first-time sires demonstrated significantly low parental scores: 0.9 ± 0.1, 1.1 ± 0.5 and 0.6 ± 0.3 for trials 1, 2, and 3, respectively (*n* = 10). Two-tailed Student *t*-tests indicated a significant difference between CD38^+/+^ and CD38^−/−^ sires in each trial (*p* < 0.01).

Based on the retrieval behavior of dams and sires shown in Figure 1, we tested OT and its analogs on parental behavior using only the sires because the capacity of the parental behavior of the dams displayed ceiling effects. As a control experiment, 30 min after a single i.p. injection of PBS, CD38^+/+^ sires showed retrieval behavior with an average parental score of 8.2 ± 0.75 (*n* = 13), while that in CD38^−/−^ sires was 1.0 ± 0.53 (*n* = 13) (Figure 2A). The parental scores of CD38^−/−^ sires 30 min after a single intraperitoneal injection of 100 ng (1 mL) per 100 g of body weight of OT, LOT-1, LOT-2, or LOT-3 were 5.4 ± 1.3 (*n* = 8), 4.1 ± 1.6 (*n* = 8), 8.9 ± 0.48 (*n* = 8) and 8.4 ± 0.56 (*n* = 8), respectively (Figure 2A). A one-way ANOVA analysis revealed significant differences: *F*_5,52_ = 13.32, *p* < 0.0001. Bonferroni’s post hoc tests showed significant differences in CD38^−/−^ sires treated with OT (*p* = 0.0158), LOT-2 (*p* = 0.001), and LOT-3 (*p* = 0.001) from CD38^−/−^ sires treated with PBS; however, no significance was observed in those treated with LOT-1 (*p* = 0.247).

The long-term effects of OT and the LOTs were examined using the same paradigm but were observed 24 h after the injections. As shown in Figure 2B, in CD38^−/−^ sires, the parental scores for OT, LOT-1, LOT-2, and LOT-3 were 1.6 ± 1.2 (*n* = 8), 6.5 ± 1.5 (*n* = 8), 4.3 ± 1.6 (*n* = 8) and 3.0 ± 1.5 (*n* = 8), respectively. A one-way ANOVA revealed significant differences: *F*_4,40_ = 3.42, *p* = 0.017. Bonferroni’s post hoc tests revealed a significant difference between the saline- and LOT-1-treated sires (*p* = 0.015), whereas no significant differences were revealed for the other treatment groups: OT (*p* = 0.999), LOT-2 (*p* = 0.515), and LOT-3 (*p* = 0.999). These data clearly show that CD38^−/−^ sires had impaired paternal retrieval behavior and that their impaired behavior was restored by OT, LOT-2 and LOT-3, but not LOT-1, just after their injection. Conversely, the long-lasting effect was obtained by LOT-1, but not by the others. These results are in accord with previous results using latency in retrieval as the basis for the behavioral scales [36,50].

### 3.2. Rapid Tests of Social Acquisition and Recognition

In the experiments on social acquisition and recognition, we examined the investigation time of resident (adult virgin male) mice to intruder virgin males for 1 min, as reported previously [8]. At 30 min after treatment with PBS or OT, CD38^+/+^ mice displayed a significant decrease at the 4th meeting with the same (familiar) target mouse from the first meeting (38 ± 1.7 s (*n* = 10) from 49 ± 1.6, *p* < 0.001 and 38 ± 1.7 s (*n* = 10) from 49 ± 1.2 s, *p* < 0.001, respectively). They also exhibited a significant increase in social contact to a novel (unfamiliar) male at the 5th meeting: 50 ± 1.0 s (*n* = 10) for CD38^+/+^ mice with PBS (*p* < 0.001) and 48 ± 1.2 s (*n* = 10) (*p* < 0.001) for CD38^+/+^ mice with OT.

Under the same paradigm, CD38^−/−^ male mice treated with PBS demonstrated a slight decrease during repeated meetings with familiar males (from 41 ± 1.7 s to 36 ± 1.7 s, *p* < 0.06). Surprisingly, at the 5th meeting with a new male, no increase in the investigation time was recorded: 32 ± 1.9 (*n* = 15) from 33 ± 1.8 s (*n* = 15) at the 4th meeting. When CD38^−/−^ male mice were treated with OT, however, they displayed a significant decrease at the 4th meeting (29 ± 1.3 s, *p* < 0.001) compared with the first meeting and a significantly longer investigation time at the 5th meeting (39 ± 1.6 s, *p* < 0.02) (Figure 3A).

The effects of the LOTs on the investigation time were examined. As shown in Figure 3B, 30 min after treatment with LOT-1, LOT-2 and LOT-3, CD38^−/−^ males displayed a decrease in interaction time with the repeated meetings: from 36 ± 3.7 s to 27 ± 1.4 s (*n* = 6) for LOT-1; from 39 ± 3.3 s to 32 ± 3.2 s (*n* = 5) for LOT-2; and from 44 ± 2.5 s to 28 ± 1.9 s (*n* = 5) for LOT-3. Significant increases in investigation time were observed at the 5th meeting with a new male from the 4th meeting with a familiar male: from 27 ± 1.4 to 36 ± 3.3 s (*n* = 6) for LOT-1; from 32 ± 3.2 to 36 ± 4.0 s (*n* = 5) for LOT-2; and from 28 ± 1.9 to 42 ± 6.0 s (*n* = 5) for LOT-3.

The average decrease following the repeated meetings from the first to the 4th meeting is illustrated in Figure 3C. The decrease was approximately 20%, except for the case of CD38^−/−^ males treated with LOT-3. A one-way ANOVA resulted showed a significant difference: *F*_6,49_ = 2.779, *p* = 0.0209. Bonferroni’s post hoc tests revealed a significant difference between CD38^−/−^ mice treated with PBS (20 ± 3.7%, *n* = 15) and with LOT-3 (45 ± 7.5%, *p* = 0.0197, *n* = 5).

Quantitative data on the percent changes between the 4th and 5th session demonstrated that only CD38^−/−^ males treated with PBS did not exhibit an increase in response to the new male (−5.1 ± 3.7%, *n* = 15; Figure 3D); all other groups displayed increases in interaction. A one-way ANOVA revealed a significant difference: *F*_6,49_ = 6.219, *p* = 0.0001. Bonferroni’s post hoc tests revealed significant differences in CD38^+/+^ males treated with PBS (23.7 ± 3.1%, *n* = 10, *p* = 0.0041), in wild-type mice treated by OT (21.8 ± 2.9% *n* = 10, *p* = 0.0096), and in CD38^−/−^ treated by OT (32.4 ± 5.6%, *n* = 5, *p* = 0.0028), LOT-2 (28.7 ± 9.7%, *n* = 5, *p* = 0.0103), and LOT-3 (34.6 ± 16.5%, *n* = 5, *p* = 0.0013) compared to CD38^−/−^ males treated with PBS. It is worth noting that no significance was observed in CD38^−/−^ males treated with LOT-1 (20.4 ± 9.4%, *n* = 6, *p* = 0.085), suggesting no immediate effect by LOT-1, as shown in Figure 1 and Figure 2.

### 3.3. Tail Suspension Test

The effects of the LOTs were examined on depression-like behavior by the tail suspension test 30 min after the injection. There was no difference in time immobility between PBS (144 ± 9.9 s, *n* = 15) or OT (159 ± 7.2 s, *n* = 5) treatments in CD38^+/+^ male mice. In contrast, the time of immobility was significantly lower in CD38^−/−^ male mice treated with PBS (76.4 ± 9.1 s, *n* = 18; Figure 4A). Immobility in the CD38^−/−^ male mice treated by OT or LOT-2, however, was significantly increased to 158 ± 25 s (*n* = 5) and 145 ± 27 s (*n* = 6), respectively. A one-way ANOVA revealed and significant difference: *F*_6,54_ = 5.922, *p* = 0.0001. Post hoc comparisons by the Bonferroni’s tests revealed significance for CD38^+/+^ with PBS (*p* = 0.0008) or OT (*p* = 0.009) and for CD38^−/−^ mice treated with OT (*p* = 0.021) and LOT-2 (*p* = 0.033), when compared with CD38^−/−^ mice treated with PBS. Statistically, no significance was observed for CD38^−/−^ mice treated with LOT-1 (88.2 ± 18 s, *n* = 6, *p* = 0.99) or LOT-3 (135 ± 20 s, *n* = 6, *p* = 0.123).

Similar findings were observed in the female mice (Figure 4B). A one-way ANOVA indicated a significant difference (*F*_6,38_ = 7.504, *p* = 0.0001). Bonferroni’s post hoc tests revealed significance in CD38^+/+^ female mice treated by PBS (169 ± 8.7 s, *n* = 10, *p* = 0.0001) or OT (167 ± 34 s, *n* = 5 *p* = 0.0011) and CD38^−/−^ female mice treated with OT (138 ± 34 s, *n* = 5, *p* = 0.034) or LOT-2 (136 ± 16 s, *n* = 5, *p* = 0.043), when compared to CD38^−/−^ mice (53.7 ± 10, *n* = 10). Similar to male mice, treatment with LOT-1 (75.4 ± 18 s, *n* = 5, *p* = 0.99) or LOT-3 (115 ± 14 s, *n* = 5, *p* = 0.375) did not increase immobility time. These data suggest that CD38 knockout adult male and female mice are less depressive, and this phenotype was recovered by OT or LOT-2 in a short time after the injection.

### 3.4. Elevated plus Maze

In the elevated plus maze task, both CD38^+/+^ and CD38^−/−^ male mice treated with PBS demonstrated a similar time spent in the open arms (Figure 5A) and distance travelled (Figure 5B): 42.8 ± 10.9 s and 0.93 ± 0.26 m (*n* = 10), respectively, for the wild-type; 27.5 ± 9.1 s and 0.66 ± 0.25 m (*n* = 5) for CD38^−/−^ mice, respectively. OT slightly increased the time spent and distance travelled in the open arms in both genotypes without statistical significance. A one-way ANOVA did not indicate significance in time spent in open arms (*F*_3,21_ = 0.542, *p* = 0.23) nor distance travelled in the open arms (*F*_3,21_ = 0.557, *p* = 0.649). These data indicate that CD38^−/−^ mice display less anxiety-like behavior, as has been shown previously [8]. We did not further test the LOTs for this parameter.

### 3.5. Sucrose Preference

Finally, we examined sucrose preference to quantify dysfunction in the sensory system and anhedonia effects as a component of the function of the nucleus accumbens. We used 1% sucrose solution in this task, according to a previous observation on CD38^+/+^ and CD38^−/−^ male mice during free-choice between water and sucrose solutions [50]. We observed no difference in consumption preference between CD38^+/+^ and CD38^−/−^ female mice treated with PBS or OT (Figure 6A), as a one-way ANOVA indicated no significant difference (*F*_3,40_ = 0.559, *p* = 0.645). In sharp contrast, as previously reported [50], CD38^−/−^ male mice showed significantly lower sucrose consumption (0.46 ± 0.04, *n* = 7) compared with CD38^+/+^ mice under the PBS injection (control; 0.78 ± 0.35, *n* = 10, *p* < 0.01) or OT administration (0.77 ± 0.03, *p* < 0.01). A one-way ANOVA revealed significance (*F*_3,30_ = 12.91, *p* = 0.0001). The phenotype in CD38^−/−^ was significantly recovered by OT administration (0.69 ± 0.05, *n* = 7, *p* < 0.05; Figure 6B).

At 30 min after the administration of the LOTs (Figure 6C), only LOT-2 demonstrated an elevation in sucrose consumption in CD38^−/−^ male mice (0.69 ± 0.05, *n* = 5, *p* < 0.05), while LOT-1 (0.58 ± 0.05, *n* = 5, *p* = 0.99) and LOT-3 (0.53 ± 0.04, *n* = 5, *p* = 0.99) were ineffective. A one-way ANOVA indicated an effect for treatment (*F*_5,33_ = 8.293, *p* = 0.0001).

Twenty-four hours after the injection, LOT-1 demonstrated an elevation in sucrose consumption (0.69 ± 0.06, *n* = 5, *p* = 0.04), with a tendency towards a long-lasting effect (Figure 6D). Mice treated with OT (0.43 ± 0.04, *n* = 5), LOT-2 (0.53 ± 0.06, *n* = 5) or LOT-3 (0.42 ± 0.04, *n* = 5) demonstrated the same level of sucrose preference as treated by PBS. A one-way ANOVA indicated a difference for treatment conditions (*F*_5,31_ = 12.40, *p* = 0.0001).

### 3.6. Pup Locomotion Test

Increased pup locomotor activity was also a phenotype that was affected by CD38 gene deletion [41]. The recovery of this phenotype by OT has not been tested previously. Therefore, we tested the effect of OT and the LOTs for the first time in the present study. After maternal separation, 5-day-old CD38^−/−^ pups treated with PBS demonstrated an elevation in locomotor activity (7.5 ± 0.3, *n* = 25) in the grid crossing test, when compared with CD38^+/+^ pups of the same age treated with PBS (3.7 ± 0.7, *n* = 25, *p* = 0.0001) (Figure 7). The locomotor activities 30 min after the injection of OT, LOT-1, LOT-2 and LOT-3 were 4.2 ± 0.6, 5.1 ± 0.8, 2.9 ± 0.8 and 2.5 ± 0.5, respectively (*n* = 10 for every cases). A one-way ANOVA indicated significance (*F*_9,125_ = 6.047, *p* = 0.0001). Bonferroni’ post hoc tests revealed significance for OT (*p* = 0.03), LOT-2 (*p* = 0.0009) or LOT-3 (*p* = 0.0002), but no significance for LOT-1 (*p* = 0.99). These data indicate that LOT-2 and LOT-3 restored the knockout phenotype better than OT, while LOT-1 had no effect.

## 4. Discussion

Here, we demonstrated the effects of OT and three different OT analogs, LOT-1, LOT-2 and LOT-3, on five types of social behaviors and locomotor activity in pups. As listed in Table 1, CD38^−/−^ mice displayed impairments in parental retrieval behavior (Figure 1 and Figure 2), social recognition (Figure 3), depression-like behavior (Figure 4) and adhedonic behavior (Figure 6), but not in anxiety-like behavior (Figure 5). Thirty minutes after OT administration to CD38^−/−^ mice, OT elicited the recovery from such deficits. As expected, LOT-1 had essentially no recovery reaction, while LOT-2 and LOT-3 showed some effectiveness. Instead, LOT-1 exhibited effects on parental behavior 24 h after the injection, which is also expected from our previous reports [35,36] (Table 1). Using novel indicators in several social behavioral tests, we confirmed the benefit of OT analogs over natural and/or synthetic OT.

In a previous report [36], we used latency to retrieve pups and total time for retrieval of 5 pups to the nest to reflect parental behavior. In the current experiment, we set the parental score. As described in the Methods, one sire or dam was rated by scores of 0, 1 and 2. Thus, it can be said that “behavioral scores” reflect the quality of retrieval behavior, which may be possible to characterize partial retrieval. The current judgement by “scores”, thus, seems to offer a more accurate evaluation of OT and LOT analogs than the “latency”.

Social recognition was tested by repeated encounters with the familiar mouse of the same sex, which differed from the previous report, in which sires investigated female virgin intruders [8,51]. The difference between the 1st and 4th encounter was not obvious in CD38^+/+^ and CD38^−/−^ mice in the current experiment, probably because of the same sex. To test that the decrease in investigation time following repeated encounters is not owing to habituation, however, the investigation duration at the 5th encounter with a novel (unfamiliar) mouse was observed. In this test, we were clearly able to discriminate between the CD38^+/+^ and CD38^−/−^ character. Through this, we could show the differential effectiveness between LOT-1 and others, such as OT, LOT-2 or LOT-3. This may lead to a potential usage of LOT-2 and/or LOT-3 individually or in combination with LOT-1 in future.

The tail suspension test is commonly used for observation of depression-like behavior. Interestingly, we discovered a shorter duration of immobility in both female and male CD38^−/−^ mice than in CD38^+/+^ mice. This may not be surprising because CD38^−/−^ mice have been shown to be hyperactive [41,52]. Recovery to the control levels were observed following OT and LOT-2 administration and, slightly, by LOT-3. As expected, no recovery was observed by LOT-1 during the short period after injection. Future experiments should investigate this after 24 h.

The sucrose preference test is usually used for investigating of anhedonia and reward processes [53]. In our experiment, female mice of both genotypes did not display any changes in sucrose preference, as reported previously [50]; however, male CD38^−/−^ mice showed less preference, and this gender-specific phenotype was clearly recovered by OT. At 30 min after the injection, only LOT-2 could replicate the OT effect. At 24 h later, OT, LOT-2 and LOT-3 were ineffective, as expected, while we observed effectiveness with LOT-1.

The mechanism how LOT-2 can show overall OT-like effects for short-time period. One of possible mechanisms may be high permeability through BBB, because of lipidation [34], although permeability was not examined yet. An alternative explanation can be made by an intensified OT induced OT release autoregulation [54,55]. The last possible mechanism of changing in social behavior is owing to activation of V1A (vasopressin) receptors [56]. However, this possibility is unlikely.

## 5. Conclusions

Here, we tested the ability of OT or LOTs to elicit the recovery of known deficits in social behaviors in CD38^−/−^ mice, which had not been examined previously. We confirmed that the lipidation strategy of adding palmitoyl acid improves the drug-like properties of OT, a peptide drug with long-lasting activity. Although the precise mechanism is unknown, these prodrug compounds slowly release side chains and metabolize to OT in the blood stream or in the brain. The next step is to investigate the pharmacodynamics of LOTs. Finally, based on these results together with previous ones, we conclude that LOTs- are beneficial over the native OT and potentially useful candidates for psychiatric diseases, including autism spectrum disorder [56].

## Figures and Tables

**Figure 1 brainsci-07-00132-f001:**
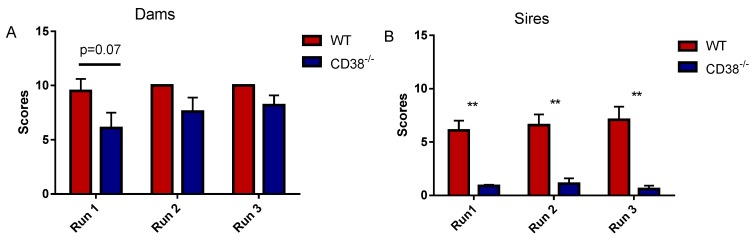
The pup retrieval scores of CD38^+/+^ Wild-type(WT) or CD38^−/−^ dams (**A**) and sires (**B**) in three trials. Ten experiments were conducted for each test. The two-tailed Student *t*-test was performed for each trial for dams (*p* > 0.05 for each comparison between genotypes) and for sires (** *p* < 0.01 for each comparison).

**Figure 2 brainsci-07-00132-f002:**
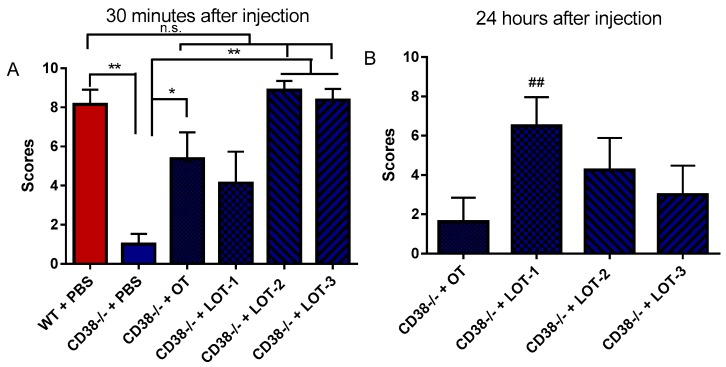
Pup retrieval behavior by the sires. The parental scores for CD38^+/+^ Wild-type (WT) or CD38^−/−^ sires at 30 min (**A**) and 24 h (**B**) after a single i.p. injection of Phosphate Buffered Saline (PBS), Oxytocin (OT), or the Lipo-oxytocins (LOTs) (100 ng/100 g of body weight); *n* = 8–10 for each test. One-way ANOVA followed by Bonferroni’s post hoc test was performed for 30 min (*F*_5,52_ = 13.32, *p* = 0.0001) and 24 h (*F*_4,40_ = 3.42 *p* = 0.017). Bonferroni post hoc test compared with CD38^−/−^ treated by PBS revealed significance at * *p* < 0.05 and ** *p* < 0.01; n.s. not significant. ^##^
*p* < 0.01 for CD38^−/−^ mice treated with OT.

**Figure 3 brainsci-07-00132-f003:**
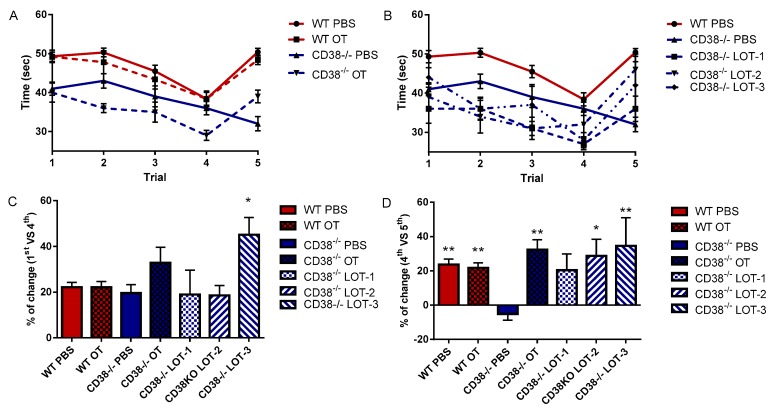
Rapid test of social acquisition and recognition. The time spent for social investigation in each trial for 1 min by CD38^+/+^ (WT) or CD38^−/−^ adult virgin males to a virgin male intruder at 30 min after a single i.p. injection of PBS, OT (**A**) Lipo-oxytocin-1 (LOT-1), Lipo-oxytocin-2 (LOT-2), or Lipo-oxytocin-3 (LOT-3) (**B**). The percent changes are illustrated between the 1st and 4th trials (**C**) and the 4th and 5th trials (**D**). *n* = 5–15 for each test. One-way ANOVA followed Bonferroni’s post hoc test revealed significance in C (*F*_6,49_ = 2.779, *p* = 0.0209) and D (*F*_6,49_ = 6.219, *p* = 0.0001). * *p* < 0.05, ** *p* < 0.01, compared with CD38^−/−^ treated with PBS.

**Figure 4 brainsci-07-00132-f004:**
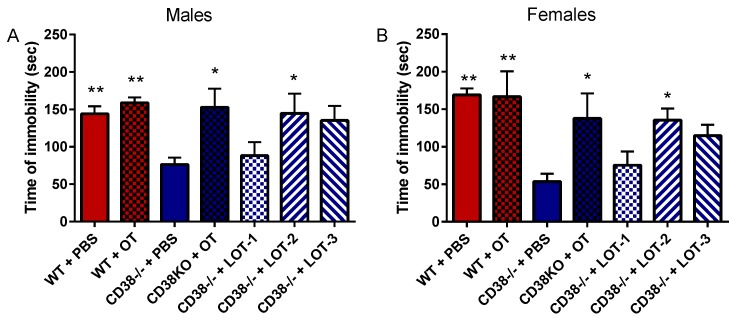
Tail suspension test. The immobility time by male (**A**) and female (**B**) CD38^+/+^ (WT) or CD38^−/−^ mice at 30 min after single i.p. injections of PBS, OT, LOT-1, LOT-2, or LOT-3. One-way ANOVA followed by Bonferroni’s post hoc test showed significance in the males (*F*_6,54_ = 5.922, *p* = 0.0001) and females (*F*_6,38_ = 7.504, *p* = 0.0001). * *p* < 0.05, ** *p* < 0.01, compared with CD38^−/−^ mice treated with PBS.

**Figure 5 brainsci-07-00132-f005:**
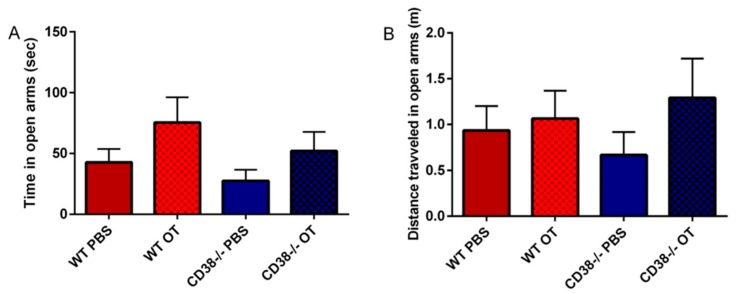
Elevated plus maze test. The time in open arms (**A**) and distance travelled in open arms (**B**) for CD38^+/+^ (WT) and CD38^−/−^ mice at 30 min after PBS or OT (single i.p. injection). One-way ANOVA revealed no significance for A (*F*_3,21_ = 1.542, *p* = 0.379) and B (*F*_3,21_ = 0.557, *p* = 0.649). *N* = 10 for each group.

**Figure 6 brainsci-07-00132-f006:**
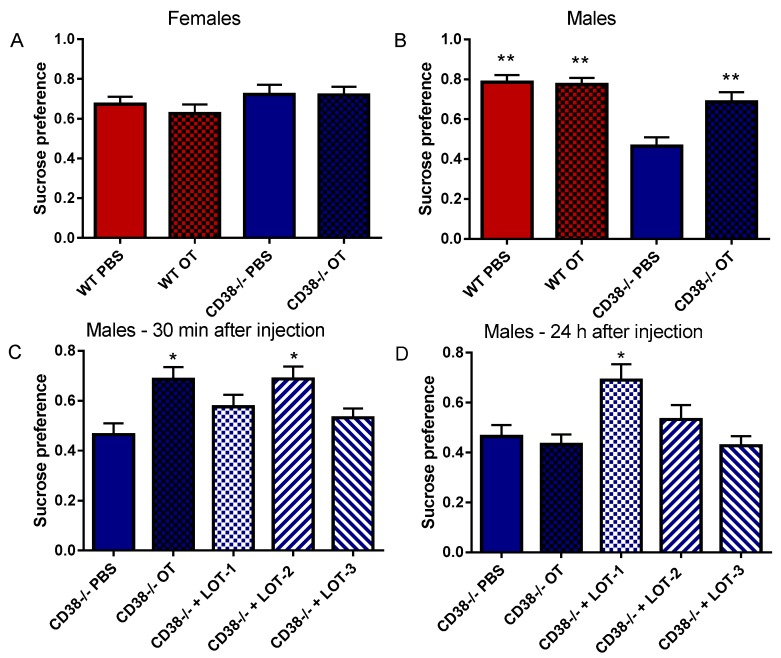
Sucrose preference test. The preference for 1% sucrose solution consumption by female (**A**) or male (**B**) CD38^+/+^ (WT) and CD38^−/−^ mice 30 min after treatment with PBS or OT (single i.p. injection). One-way ANOVA was performed and revealed no significance for A (*F*_3,40_ = 0.559, *p* = 0.645) but significance for B (*F*_3,30_ = 12.91, *p* = 0.0001). ** *p* < 0.01 from CD38^−/−^ with PBS. Sucrose preference in males treated by PBS, OT, or the LOTs 30 min (**C**) or 24 h after injection (**D**). A one-way ANOVA followed by Bonferroni’s test revealed significance at 30 min after injection (*F*_5,33_ = 8.293, *p* = 0.0001) and 24 h after injection (*F*_5,31_ = 12.40, *p* = 0.0001). * indicates *p* < 0.05, ** indicates *p* < 0.01, compared with CD38^−/−^ male mice treated with PBS.

**Figure 7 brainsci-07-00132-f007:**
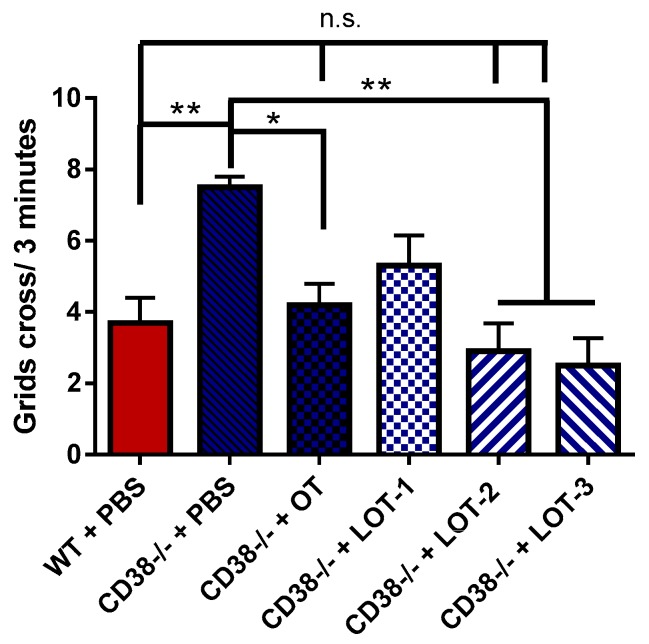
Locomotor activity in mouse pups. Pups of 38^+/+^ (WT) and CD38^−/−^ mice were subjected to the grid crossing test 30 min after a single i.p. injection of PBS, OT, LOT-1, LOT-2, or LOT-3. A one-way ANOVA followed Bonferroni’s post hoc test was performed (*F*_9,125_ = 6.047, *p* = 0.0001). * *p* < 0.05, ** *p* < 0.01 in each comparison. n.s., not significant.

**Table 1 brainsci-07-00132-t001:** Summary of the behavioral profiles and effects of Oxytocin (OT) and the Lipo-oxytocins (LOTs) on deficits in CD38 knockout(^−/−^)mice. The arrows indicate increases or decreases in the phenotypes in CD38^−/−^ mice compared with those in CD38^+/+^ mice. ++ or + indicates recovery; 0 represents no effect. OT, oxytocin; LOT, lipo-oxytocin; n.t., not tested.

**30 min after Injection**	**CD38^−/−^**	**+OT**	**+LOT-1**	**+LOT-2**	**+LOT-3**
Parental scores	↓	+	0	++	++
Social recognition	↓	++	0	+	++
Depression-like behavior	↓	+	0	+	0
Anxiety-like behavior	0	0	n.t.	n.t.	n.t.
Adhedonic	↓	+	0	+	0
Pup locomotion	↑	+	0	++	++
**24 h after Injection**	**CD38^−/−^**	**+OT**	**+LOT-1**	**+LOT-2**	**+LOT-3**
Parental scores	↓	0	++	0	0
Adhedonic	↓	0	+	0	0

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
