# Peer review of "Effects of Three Lipidated Oxytocin Analogs on Behavioral Deficits in CD38 Knockout Mice"

_brainsci, 2017, doi:10.3390/brainsci7100132_

Round 1

Reviewer 1 Report

Comments for Authors

In the manuscript entitled “Effects of Three Lipidated Oxytocin Analogs on 3 Behavioral Deficits in CD38 Knockout Mice” by Cherepanov,S. et al described the different effects of three lipidated oxytocin analogue on  five type of social behaviors, and additional locomoter activity in pups, using wt and CD38-/- mice, probably chosen as a model disease animal related to ASD. They performed pharmacological evaluation against three different lipidated oxytocin analogues, in comparison with authentic OT, by five types of social behaviors and additional locomoter activity tests.

They described delayed and potentially long-lasting activity of LOT-1, in comparison with OT, LOT-2 and LOT-3, especially by pup retrieval behavior test by the sire, 24 hours after injection of those compounds.

Overall, the approach is appropriate and the result is interesting, except a couple of flaws to be checked and revised.

1.      Though the all behavioral results were demonstrated clearly in each figures, the law and detailed data were listed in sentences, and aren’t easy to read. Instead, detailed data including (n) value should be listed in tables (supplement). 

2.      Although, the primary finding of authors might be long-lasting activity of LOT-1, authors didn’t describe the effects observed 24 hours after LOT-1 administration to CD38-/- mice, in behavioral test of social recognition, depression like behaviors, and pup locomotion. In case of depression–like behavior test 24 hrs after injection, might be omitted.  

3.      Authors used separately “30 minutes after treatment with LOT-1” in line 246 and so on, and “at 30 min after single i.p injection of PBS, OT…” in line 241 and so on. If the treatment means other than i.p injection, author should describe this in materials and methods. If they are the same, please unify those to avoid confusion.  

4.      line 68, remove “it”

5.      line 78, remove “it”

6.      line 145, able contact -> able to contact

7.      line 247, “sec and to 27…”, consider “and”

Author Response

Dear Reviewer, thank you very much for your qualified report.
 we uploaded revised manuscript with added suppliment 

Response for comments
1. "Though the all behavioral results were demonstrated clearly in each figures, the law and detailed data were listed in sentences, and aren’t easy to read. Instead, detailed data including (n) value should be listed in tables (supplement). "

1.  This comment is reasonable. We prepared tables contain detailed information about means, SEM, n and p value. Tables included to the supplement.

2.     Although, the primary finding of authors might be long-lasting activity of LOT-1, authors didn’t describe the effects observed 24 hours after LOT-1 administration to CD38-/- mice, in behavioral test of social recognition, depression like behaviors, and pup locomotion. In case of depression–like behavior test 24 hrs after injection, might be omitted.  

2.  We appreciated this comment. Social recognition, pups locomotion test 24 hrs after injection will be performed in near future.

3. Authors used separately “30 minutes after treatment with LOT-1” in line 246 and so on, and “at 30 min after single i.p injection of PBS, OT…” in line 241 and so on. If the treatment means other than i.p injection, author should describe this in materials and methods. If they are the same, please unify those to avoid confusion.

3. This is important point. It is the actually same i.p. treatment.   

4. line 68, remove “it”

4. Fixed

5.     line 78, remove “it”

5. Fixed

6. line 145, able contact -> able to contact

6. Fixed (now fragment in line 150)

7.   line 247, “sec and to 27…”, consider “and”

7. Fixed (now fragment in line 250)

Reviewer 2 Report

This is a well written, carefully prepared manuscript. The limited aims of demonstrating effects of administrating lipidated-oxytocin peptides on behavior in CD38-/- mice are well-done and convincing.

The only sentence that needs reworking is line 77-78."LOT-2 and LOT-3 had minor effects on parental behavior after 30 minutes, but it they had some effects after 3-12 hours that were not as evident as LOT-1 [36]". It is clunky and I'm not sure what the authors are saying. Please fix.

My major overall concern with these studies is an apparent paradox: In the introduction (71-81), the authors state the LOT's have no direct reaction of stimulating OT receptors, no increase in oxytocin levels at 30 min, and no or minor effects on behavior at 30 min after administration. Nonetheless, most of the studies in this report are done 30 min after peptide treatment. The reader is left without a plausible mechanism for the effects of these compounds on behavior after only 30 min treatment. A paragraph in the discussion addressing these issues would be most helpful.

Author Response

Dear Reviewer, thank you very much for your qualified report.
we uploaded revised manuscript with added suppliment 

Response for comments.

The only sentence that needs reworking is line 77-78."LOT-2 and LOT-3 had minor effects on parental behavior after 30 minutes, but it they had some effects after 3-12 hours that were not as evident as LOT-1 [36]". It is clunky and I'm not sure what the authors are saying. Please fix

     1. We agree with this comment. Because LOT-2 and LOT-3 shown ability to recover one parameter of parental behavior, but fail for another. While in case of LOT-1 it is clearer. We changed sentence in manuscript See line 77-82

    2. My major overall concern with these studies is an apparent paradox: In the introduction (71-81), the authors state the LOT's have no direct reaction of stimulating OT receptors, no increase in oxytocin levels at 30 min, and no or minor effects on behavior at 30 min after administration. Nonetheless, most of the studies in this report are done 30 min after peptide treatment. The reader is left without a plausible mechanism for the effects of these compounds on behavior after only 30 min treatment. A paragraph in the discussion addressing these issues would be most helpful

    2. We appreciated your intellectual comment. This point must be improved. LOT-1 demonstrates absence of effect on calcium concentration elevations, while LOT-3 demonstrated marginal effect and LOT-2 demonstrated activity like a  week partial agonist with an EC50 lower than that of OT by 1000 times. Though very low, the order of direct activity was LOT-2>>LOT-3>>>LOT-1. LOT-2 demonstrated strong effects in 30 min in all tests. LOT-2 was not active in the tail suspension test and sucrose preference test. And LOT-1 failed to show any effect in all tests 30 min after injection. These results may be reflected by in vitro effects in some content.

Another possible mechanism was added to discussion, like higher BBB permeability, OT-induced OT autoregulation and V1A receptors activation.

These issues were added to introduction (line 83-86) and discussion (line .402-406)

Reviewer 3 Report

The focus of this paper was to determine if palmitoyl modified derivatives of the peptide oxytocin (i.e. LOT-1, LOT-2, LOT-3) can show both favorable and enhanced effectiveness in the behavioral outputs of CD38-/- mice.  These mice express autism like characteristics including behavioral deficits in social interactions, depressive like behaviors, and diminished reward tendency.  The results show temporal effects of the LOT-1 derivative, with beneficial social effects occurring  at 24 hours post drug treatments, with no acute effect of the drug.  LOT-2 and LOT-3 derivatives did not alter behavioral changes long term, however they significantly improved behavioral outcomes greater than unmodified oxytocin.  Furthermore, the manuscript described an alternative method to score parental retrieval behavior.  Overall the manuscript was well written, with an array of interesting results.  A few minor points may be addressed in the discussion/conclusion to further add upon this manuscript. 

For discussion/conclusion, LOT-1 is focused on a lot for its effectiveness 24hrs post treatment.  While intriguing, I feel like it is a bit too focused on this compound, as there were only 2 parameters examined at a 24 hr timepoint.  However, the authors did suggest that further studies into this drug would be enacted.

The effects with LOT-2 and LOT-3 are interesting, as they both seem to have enhanced effects over regular oxytocin in some cases.  More discussion into this effect would be beneficial.

Furthermore, have any studies combining LOT-1 with LOT-2/3 been proposed?  It would be nice to see both an immediate and prolonged effect.

Table 1.  Is this the best way to represent/summarize the data?  It seems to be combining the effects at 30 minutes and 24 hrs for all of the LOTS.  This can be misleading to the reader.

Author Response

Dear Reviewer, thank you very much for your qualified report.
we uploaded revised manuscript with added suppliment

Response for comments.

 For discussion/conclusion, LOT-1 is focused on a lot for its effectiveness 24hrs post treatment.  While intriguing, I feel like it is a bit too focused on this compound, as there were only 2 parameters examined at a 24 hr timepoint.  However, the authors did suggest that further studies into this drug would be enacted

    1. The comment is quite reasonable. We amended old Conclusion. In new Conclusion, we mentioned not too much of LOT-1, but LOTs. Lines 414-415

    2. The effects with LOT-2 and LOT-3 are interesting, as they both seem to have enhanced effects over regular oxytocin in some cases.  More discussion into this effect would be beneficial.

    2. We appreciated this comment. We discussed the possible mechanisms in lines 402-406

    3. Furthermore, have any studies combining LOT-1 with LOT-2/3 been proposed?  It would be nice to see both an immediate and prolonged effect.

    3. It is an attractive idea to use different benefits of analogs. We mention it  (lines  387-389)

    4.Table 1.  Is this the best way to represent/summarize the data?  It seems to be combining the effects at 30 minutes and 24 hrs for all of the LOTS. This can be misleading to the reader.

    4. We agree with this comment. We edited the summary table with highlighting of difference between 30 minutes and 24 hours for OT and LOTs. Please check new version between lines 373 and 374.
